# Collaborative Learning through a Virtual Community of Practice in Dementia Care Support: A Scoping Review

**DOI:** 10.3390/healthcare11050692

**Published:** 2023-02-26

**Authors:** Justice Dey-Seshie Dedzoe, Agneta Malmgren Fänge, Jonas Christensen, Connie Lethin

**Affiliations:** 1Department of Health Sciences, Lund University, 221 00 Lund, Sweden; 2Department of Social Work, Malmö University, 205 06 Malmö, Sweden

**Keywords:** dementia, formal caregivers, informal caregivers, reflective collaborative learning, resilience capacity, virtual community of practice

## Abstract

The aim of this scoping review was to identify, synthesize, and report research on reflective collaborative learning through virtual communities of practice (vCoP), which, to our knowledge, is scarce. A second aim was to identify, synthesize, and report research on the facilitators and barriers associated with resilience capacity and knowledge acquisition through vCoP. The literature was searched in PsycINFO, CINAHL, Medline, EMBASE, Scopus, and Web of Science. The Preferred Reporting Items for Systematic Reviews (PRISMA) and Meta-Analyses for Scoping Reviews (ScR) framework guided the review. Ten studies were included in the review, seven quantitative and three qualitative studies, written in English and published from January 2017 to February 2022. The data were synthesized using a numerical descriptive summary and qualitative thematic analysis. Two themes: ‘knowledge acquisition’ and ‘strengthening resilience capacity’ emerged. The literature synthesis provides evidence of a vCoP as a digital space that supports knowledge acquisition and strengthens resilience for persons with dementia, and their informal and formal caregivers. Hence, the use of vCoP seems to be useful for dementia care support. Further studies including less developed countries are, however, needed to enable generalizability of the concept of vCoP across countries.

## 1. Introduction

Major neurocognitive disorders are often referred to as dementia diseases [1,2]. The most common forms of dementia are Alzheimer’s disease (AD) and vascular variations, with 4.2% and 1.0% global prevalence, respectively, among people aged ≥65 [2,3]. A dementia disease results in declining cognitive capacity, subsequent declining functioning [2,4], and increasing dependence in everyday activities on informal and formal caregivers [2,3,5]. As highlighted by Lethin et al. [6], close to 75% of homecare for persons with dementia is provided by informal caregivers, but, over time, the need for formal help increases. Despite the complexity of problems people with a dementia disease and their informal and formal caregivers face, there is a lack of knowledge and skills in dementia care, subsequently affecting the well-being and quality of life (QoL) of persons with dementia and their informal and formal caregivers negatively [6]. A study conducted in China [7] highlighted that more than 80% of informal caregivers to people with a dementia disease were illiterate, with none or very little basic level education. The informal caregivers in this study reported a low knowledge level in the care for persons with dementia, including a lack of interest in practicing person-centered care. Another review [8] reported that a lack of academic policy guidelines accounted for insufficient professional training, resulting in only average knowledge and competency in dementia care among formal caregivers. Increased knowledge has also been shown to relieve the negative impact when it focuses on both the informal, formal caregivers and the persons with dementia’s personal goals and development [9].

Reflective collaborative knowledge acquisition may be described as dynamic repetition of an activity for the construction of knowledge and knowledge sharing [10]. It aims to obtain group learning outcomes by involving personal knowledge-building as part of the process, including tacit and precise knowledge. Credited to Lave and Wenger [11], the Community of Practice (CoP) concept evolved from the situated nature of learning. It fosters collaborative learning to achieve goals set over, for example, digital platforms, and has been applied in various professional fields to include education, among others [12,13,14,15]. Participation in and the sustainability of a CoP is very much dependent on the individuals finding the value of and need for it [16]. 

More recently, the use of technology in healthcare has been integrated into the CoP, and, increasingly, the virtual Community of Practice (vCoP) [16,17]. This is consistent with the assistive aid envisaged by the European Commission (EC) [18]. Contrasting vCoP to other CoPs, the virtual space promotes flexibility, enhances time management, and eases and speeds up communication. This is achieved through connecting members in different geographical locations, while ensuring face-to-face interaction over videoconferencing. Thus, to mitigate the existing lack of knowledge for informal and formal caregivers, and to improve care provision and outcomes for persons with dementia, reflective collaborative knowledge acquisition through vCoP has shown the capacity to strengthen resilience [9,19], such as adapting well when facing stressors [20]. Resilience has been an integral part of public health policies and can be explained as building up adaptive, absorptive, proactive, and transformative abilities at the individual, community, and system levels [13,14,20]. Moreover, patient-level outcomes, as well as knowledge, skills, and attitudes, have been shown to improve using vCoP [16], and the use of digital technology influenced nurses’ participation in knowledge sharing within the virtual space [21]. Thus, vCoP, including formal and informal caregivers in dementia care, may be relevant when managing persons with dementia to improve their self-efficacy and QoL. Such knowledge is, however, scarce, and novel approaches to this field of study are required.

The overarching aim of this study was to identify, synthesize, and report on available evidence on reflective collaborative learning through vCoP to strengthen resilience capacity among persons with dementia diseases, as well as their informal and formal caregivers. The specific questions were: (i) How can reflective collaborative learning among people with dementia, and their informal and formal caregivers, be acquired through vCoP? (ii) What are the facilitators associated with reflective collaborative learning through vCoP among people with dementia, and their informal and formal caregivers? Finally, (iii) what are the barriers associated with strengthening resilience capacity through vCoP among people with dementia, and their informal and formal caregivers?

## 2. Materials and Methods

### 2.1. Study Design 

This study is a scoping review, guided by the PRISMA-ScR [22] framework. Scoping review methodology determines the coverage, volume, and availability of scientific evidence in the literature on a given topic [23]. Additionally, unclear emerging concepts, theories, and research gaps can be identified to inform practice [22,23].

### 2.2. Search Strategy and Databases

A literature search was conducted to identify the primary and potentially relevant journals in the following bibliographic databases: the American Psychological Association (APA) Thesaurus of Psychological Index Terms (PsycINFO); the Cumulative Index to Nursing and Allied Health Literature (CINAHL); Medline (PubMed); EMBASE; Scopus; and Web of Science. The first choice of database was influenced by its notability to provide materials on psychiatry and mental health-related topics. The second choice was noted for papers on topics related to nursing, and the third was noted as a free database for accessing references on, e.g., life science and biomedical topics [21,24,25]. 

With support from two experienced librarians, the search terms and strategies were drafted and refined. A search was conducted in CINHAL with index and non-index terms. This was repeated among the rest of the chosen electronic databases, after adjustments to the various subject headings and non-index terms, i.e., MeSH in Medline, APA Index Term in PsycINFO, and so on. The search terms were combined with the Boolean Operators in each of the databases. The search filters applied included publication years from January 2017 to February 2022, and English language. The detailed literature search conducted was guided by the population, concept, and context (PCC) framework [26]. 

### 2.3. Eligibility Criteria

To enhance the diversity of the various methods employed in the primary studies, the health-related articles with content according to the PCC framework were included [26]. Quantitative and qualitative studies from all geographical contexts and study settings, written in English and published from January 2017 to February 2022, were also included.

### 2.4. Selection of Sources of Evidence

A total of 1359 records were identified by the first (J.S.D.) and last authors (C.L.). A literature peeling added two citations to obtain 1361 for screening, while a grey literature search yielded no results. Using Covidence software (app.covidence.org) [27], the citations and screening were managed independently by each author, respectively. Following a de-duplication, 1294 records were screened for the title and abstract. A full-text review was completed for 35 records. Conflicts were resolved and consensus reached through discussions between the first and last authors. Ultimately, ten studies (quantitative and qualitative) were included in the data charting process for the review. This was completed manually by the first author to ensure that the studies included addressed the research questions/aims.

The screening and data charting form was amended before the beginning of the screening process, with support from a librarian to increase consistency after the initial screening by the authors. The potentially relevant journals identified by the search were charted independently by the authors through the reading of titles, abstracts, and full texts. The data were abstracted based on the following: the study characteristics (author, year, country, objectives, and ethics), study design, sample population, sample size, study context, concept, and results/findings (Table 1). The screening process in Covidence populated the results into the PRISMA flow chart [27]; see Figure 1. To report this scoping review, the PRISMA-ScR was used [26] (Appendix A).

### 2.5. Data Analysis

Thematic analysis is an apt qualitative analytical tool [38], and, in this review, it was used to identify, analyze, organize, describe, and report emerging themes. The analytical phase consisted of (i) presenting a numerical descriptive summary and qualitative thematic analysis, and (ii) reporting the findings through themes that align with the study aim [39]. This followed guidelines articulated by Braun and Clarke [38].

The articles were read thoroughly to start the thematic analysis process while understanding the content. The codes were generated and clustered into sub-themes by differences and key findings. The sub-themes were defined, named as key themes, and subsequently refined to meet the aims of the review. To engage more deeply, and to increase the verification of the analysis process during the coding process, a peer debriefing between the first and last authors was applied [39]. 

### 2.6. Ethical Considerations

The review considered studies with primary ethical approval with adherence to the Declaration of Helsinki [40], the General Data Protection Regulation (GDPR) [41] for European countries, as well as the principles spelt out in the Belmont Report, as articulated by Beauchamp and Childress [42] and Polit and Beck [43]. To that end, the publications included in the review were assessed to have met ethical guidelines, as suggested by Vergnes et al. [44], when conducting a scoping review.

## 3. Results

The ten included publications covered both knowledge acquisition and facilitators associated with strengthening resilience capacity.

### 3.1. Study Characteristics

The studies included were published between 2017 and 2022, and conducted in Europe (4), the USA (4), and South America (1). One study was carried out across three countries: the USA, Canada, and the UK.

The virtual media used included Facebook; a face-to-face internet survey platform; an online dementia education library; online Tele-Savvy; Webnovela Mirela; the application “Estic amb tu - I’m with you”; Massive Open Online Course (MOOC); online video consultation; a friend-sourcing Web application; and a social media blog genre, “the church of online” [28,29,30,31,32,33,34,35,36,37] (Table 1). Three studies employed qualitative research methods with a longitudinal content analysis and participatory action research design, respectively [28,29,34]. Seven studies used quantitative methods [30,31,32,33,35,36,37]. 

### 3.2. Themes

From the data analysis, two themes, including six subthemes, emerged (Table 2).

#### 3.2.1. Knowledge Acquisition

The theme ‘knowledge acquisition’, comprising knowledge sharing and knowledge development, was present in all ten studies [28,29,30,31,32,33,34,35,36,37]. The concept meant learning with and from one another through an online platform, including Facebook sites and online Tele-Savvy, to mention a few.

Knowledge sharing was described as experiences and perceptions [32,37] psychosocial well-being [29,35,37], and health literacy [28,30]. This is illustrated by those involved in a vCoP, where the experiences and perceptions shared reduced the burden, perceived stress, depressive symptoms, and BPSD remarkably [32,37]. In a Facebook discussion post and a blog site, informal caregivers and persons with dementia shared knowledge regarding psychosocial well-being (self-feeding, medication adherence, and interpersonal communication, among others) [28,29]. Similarly, on a daily basis, informal caregivers shared their knowledge regarding psychosocial well-being, such as offering guidance, suggesting solutions, providing encouragement and condolences, and empathizing with each other [35,37]. Practical support knowledge was shared by experienced formal caregivers, who had a higher health literacy rate, to informal caregivers [28,30].

Knowledge development included sharing of, for example, positive experiences [36] and psychosocial well-being activities [31,37]. Informal caregivers applied new caregiving ways to improve their specific measures, as the cognitive function and psychological flexibility in the person with dementia disease declined [31,37]. Shared experiences and perceptions among caregivers in an online Tele-Savvy psychoeducational program and a blog genre enabled them to develop knowledge and skills regarding dementia care competency [28,32]. Strategies for self-management knowledge and skills were also developed after participants watched an online Telenovela and used the Friendsourcing Web application aimed at teaching informal caregivers coping skills to handle stressful care situations, as well as emotional situations [33,37]. The informal caregivers also gained valuable knowledge regarding care support efficiency via a video consultation [34].

#### 3.2.2. Strengthening Resilience Capacity

The majority of the studies touched upon a variety of factors that acted significantly to strengthen resilience capacity. Resilience capacity was explained to mean situations that encouraged persons with dementia, as well as informal and formal caregivers, to cope with dementia care stressors [29,30,31,32,33,34,35,36]. In addition, self-esteem [29,30,31,32,35,36], gaining new knowledge [33,34], improving informal caregiver competence [32], and females as informal caregivers [36] were included in the subthemes conceptualized.

Facilitators associated with strengthened resilience capacity included developing self-esteem, and emotions such as feeling useful and positive through the action of contributing to society, played a major role for people with dementia and their informal caregivers [28,29,30,31,32,33,34,35,36]. Success and positive feelings [29], self fulfilment, and feeling more useful when caring for a significant relative with dementia [30,36] were considered an important part of self-esteem, contributing to strengthened resilience capacity. Furthermore, improvements and achievement in personal values [28,31], the provision of an enabling environment, and support programs [28,33,34,35] contributed to strengthened resilience capacity. In this context, new knowledge and skills gained and developed by participation in vCoP were reported to be essential [29,30,31,32,33,34,35,36].

Particularly, new knowledge on care support efficiency enabled informal caregivers to develop their presence, attention, concentration, connectedness, and focus [28,34] to practice knowledge acquired on a daily basis [35,37]. Informal caregiver programs in vCoP increased informal caregivers’ competence and decreased caregiver burden, stress, depression, BPSD, and caregivers’ reaction to BPSD [32,37], also contributing to strengthen resilience capacity. Furthermore, new knowledge gained by caregivers regarding technology-enabling support programs, while reaching large target groups in a cost-effective manner, contributed to building resilience capacity [33].

“Females as informal caregivers” was reported as a particular phenomenon to informal caregiving [42]. Female spouses or children of the person with dementia usually dominate in caregiving. This was evident in the sampling conducted to participate in an application “Estic amb tu - I’m with you”, an online social support network to evaluate the QoL of informal caregivers participating in vCoP. However, unfulfilled informal caregiver responsibilities were understood as situations that negatively influenced the acquisition of knowledge. In particular, challenging circumstances, including the inability to multitask or to apply a therapeutic touch among others during an online community, served as demotivation in acquiring the requisite dementia care knowledge through vCoP. The caregivers reported feeling anxious, helpless, dissatisfied, and uncomfortable during a video consultation, due to the lack of physical contact, which served as a prompt for participation [40].

## 4. Discussion

In this scoping review, ten studies were included, comprising data from Europe, the USA, and South America. Seven studies used a quantitative design, and three studies a qualitative design.

Our findings show that most participants of a vCoP acquired knowledge by listening to others sharing their knowledge and experiences. This finding confirms that effective knowledge acquisition in a vCoP is dependent on the participants’ ability to critically interpret and respond to shared information from others. It also gives room to interpret and understand the differences among individual attitudes and the type of knowledge they share to impact the larger community [45]. As emphasized by Lave and Wenger [11], the accomplishment of set goals by participants can be achieved through fostering mutual group learning processes and knowledge sharing. 

A range of emotional factors, from the perspective of informal and formal caregivers and persons with dementia, contributed immensely towards the facilitation of strengthening resilience capacity through a vCoP. These findings are similar to findings from a study examining the experience of a vCoP among instructors and students at the time of the COVID-19 pandemic [46]. Although the results of the latter study reflect experiences in another population, such experiences seem to be quite general. Learning new knowledge from other members of a vCoP has been indicated to improve individual care competency, concentration, attention, and presence, and to deal with a stressful dementia care burden while at the same time improving QoL. In dementia healthcare, these personal attributes and professional competency are essential for caregivers to manage difficult and challenging clinical and personal situations [47]. This is also in line with a previous scoping review [48] on dementia care boot camp, showing that healthcare students with prior knowledge in dementia care were significantly more confident than students with less or no dementia care knowledge.

In contrast to the factors that supported informal and formal caregivers in strengthening their resilience capacity, informal and formal caregivers were also unfulfilled, helpless, and uncomfortable with their life situation [49]. In this study, they pointed out their inability to multitask and to apply a therapeutic touch in a virtual context. Frequent changes that took place in professional guidelines were sadly communicated during the vCoP, resulting in great demotivation. A number of unorganized directives from their line managers also existed. This could be attributed to the inability of informal caregivers to set aside time for participation in the vCoP, lack of digital knowledge related to the use of digital resources, and low-level preparation among informal and formal caregivers. Similar findings were reported by Hassan [50], revealing several informal caregiver-related barriers to using technology in their routine activities, e.g., the design of the Internet platform and time constraints. Such factors should be considered when planning and conducting a vCoP for informal and formal caregivers of persons with dementia.

Developing self-esteem, gaining new knowledge, and informal caregiver competence seem to be important factors contributing to strengthening resilience capacity for caregivers caring for people with dementia. This study showed that emotions such as feeling useful and positive played a major role for both people with dementia and their informal caregivers [29,30,31,32,33,34,35,36]. A review [51], aiming to investigate what works to support informal caregivers of people with dementia found, among other things, that “finding self-development, growth, and meaningfulness in life through the care experience” were factors related to resilience. This may in some way be equivalent to feeling useful and positive to be able to develop self-esteem. Informal caregiver competence seems to increase by attending programs in vCoP regarding dementia care. The results of this study showed that when informal caregivers’ competence increased, caregiver burden, depression, BPSD, and caregiver reaction to BPSD decreased [32]. This is in line with the result of Hepburn et al. [52], who developed and tested a transportable training program for informal caregivers of people with dementia. Their results showed that informal caregivers reported increased confidence, knowledge, and skills. This makes it important that professionals and stakeholders are aware of factors contributing to strengthening resilience capacity for caregivers caring for people with dementia. 

A leading principle in the United Nations Agenda 2030 for Sustainable Development is “leaving no one behind”, seeking to realize the human rights of all, and shifting the world towards a sustainable and resilient path [53]. In Horizon Europe’s Strategic Plan 2021–2024 [54], the key strategic orientation is “Creating a more resilient, inclusive and democratic European society”. Thus, making use of innovative, inclusive, and efficient ways to contribute to lifelong learning is essential for the goal’s achievement. The use of CoPs is democratic and inclusive, and has the potential to improve learning among all citizens [55]. Subsequently, CoPs can contribute to improving public health, also, in low-income countries. 

### Strengths and Limitations

It is worth noting that the current review followed a rigorous methodology, as prescribed in the PRISMA-ScR framework [56], which enabled the essential data to be included. The aim of this review enabled the inclusion of primary studies that employed both quantitative and qualitative designs. This is deemed appropriate to identify, synthesize, and report on the various aspects of the specific objectives in the performance of the review. Nonetheless, the current scoping review was not devoid of drawbacks. The limited amount of evidence found is suggestive of lack of a broader perspective on the topic, which affects the generalizability of the findings. This may lead to the authors losing sight of other factors, such as demographics, e.g., age and geographical location, that could have a significant influence on vCoP participation, as well as the results from a vCoP. The studies included in this scoping review were restricted to the Western world, and, to open up the scope, there is a need for studies conducted in other parts of the world, such as other less developed countries.

No quality appraisal was conducted of the included studies. This may undermine the quality of the current synthesis, although the process is not mandatory for a scoping review. Finally, included in this review were publications written in English only, and no grey literature was included. This may have excluded important publications. However, given the theoretical topic targeted, we concluded that extending the inclusion criteria would have added a significant complexity to the search and review process, while, at the same time, not providing an added value. On the contrary, we consider our methodology to provide credibility to the synthesis useful to inform future research, as well as to feed into dementia care policy and practice.

## 5. Conclusions and Implications

We conclude that vCoP as platforms seem to support knowledge acquisition and strengthen resilience for persons with dementia, as well as informal and formal caregivers, locally and globally. The findings can serve as a basis for stakeholders responsible for the enactment of mental health policies globally to enhance the implementation of scientific evidence into practice using innovative and inclusive methods. For clinical practice, it is essential for dementia care providers to become aware of current knowledge and alternative, cost-effective ways of providing dementia care. Action in countries where less attention is given to people with dementia may support bridging the gap between the less developed and the developed countries regarding dementia care, toward the achievement of the Agenda 2030 and the Sustainable Development Goals, SDG. Further research in less developed countries enabling the inclusion of other perspectives, and to increase the generalizability of evidence is, however, needed.

## Figures and Tables

**Figure 1 healthcare-11-00692-f001:**
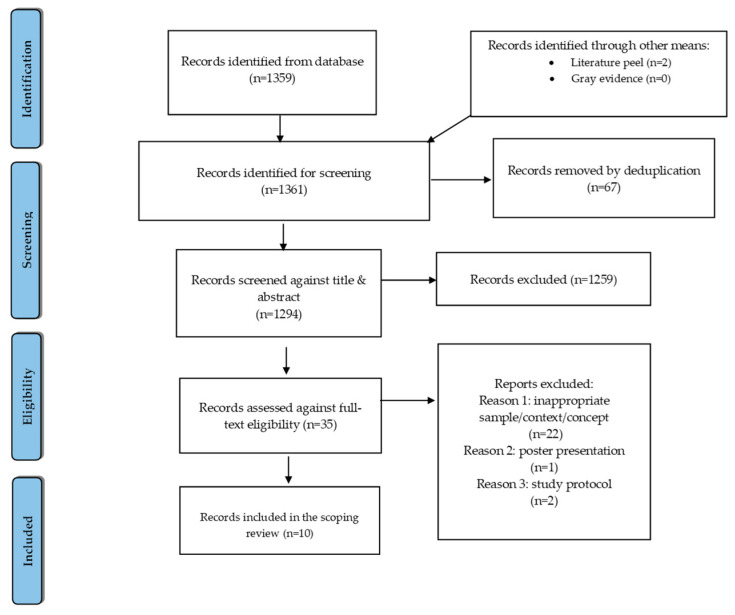
PRISMA flow chart for inclusion and exclusion of studies in this scoping review [27].

**Table 1 healthcare-11-00692-t001:** Overview of included studies in scoping review for collaborative learning acquisition through virtual Community of Practice in dementia care support.

Author, Year, Country	Aim(s), Design, Sample Population and Ethics	Study Context and Concept	Results
Anderson et al. (2017) [28], Utah, USA	Aim: The study explored how family caregivers of PwD use the social media platform (blogs) as part of the individual caregiving experience. Design: Qualitative descriptive content analysis Sample population: Family caregivers. Sample size: 10 blog sites Ethics: Yes	Context: Social media blog genre “the church of online support”. Concept: Social media blog genre on dementia caregiving experience.	The social media blog genre content analysis showed family caregivers used blogs as platform: for engaging and communicating with others: “…a place to update people…”; to seek and disseminate information; to record caregiving experience, decline of disease, etc.;“ …to document [for] whatever purpose...”; “…as a way of contributing to society…“; and “…to write notes so that later down the road I can write a manual or a book and donate the proceeds.”
Bachmann (2020) [29], USA, the UK and Canada	Aim: To examine the nature of the caregiver’s work, its mental and physical demands, experience and questions, and the relationship between the persons with Alzheimer’s dementia (AD), the caregiver, and family members. Design: A content analysis of longitudinal social media communication. Population: Persons with Alzheimer’s dementia, family caregivers, care professionals. Sample size: 28 Ethics: Yes	Context: Facebook social networking site. Concept: Facebook post on Basic Activity of Daily Living (BADLs) and Instrument Activity of Daily Living (IADLs) support information from persons with Alzheimer’s dementia, family caregivers, care professionals.	Discussion posts from patients included: self-feeding, healthcare and well-being, e.g., taking prescribed medication; positive effects of entertainment, e.g., watching movie, listening to music, etc.; transferring/moving in the community; managing money; shopping; preparing meal; getting help from family members; sleeping disorders; appointments, e.g., tests and surgeries; caregivers and facility selection; ageing, death, and dying. Others include engagements in interpersonal communication between caregivers and persons with Alzheimer’s dementia; religious observance, holiday, or birthday anniversary; social groups and other public events; care of pets; and safety procedures and emergency response. Discussion post from family caregivers and care professionals included: exhaustion and feeling of giving up; changes in health state of a patient caregivers success and positive feeling; violent behavior; financial issues.
Efthymiou et al. (2021) [30], Greece and Cyprus	Aim: Identify the levels of Health Literacy (HL) and eHealth Literacy (eHL) among carers of person with dementia in Greece and Cyprus, and to search for the associations with other caring concepts. Design: Descriptive correlational. Sample population: Informal carers of persons with dementia. Sample size: 241 Ethics: Yes.	Context: Face to face Internet survey platform. Concept: Care-giving self-efficacy, coping strategies, care-giving perceptions, and social support.	Higher score on eHeals-Carer “information seeking” was related with higher use of emotion-focused strategies, care-giving self-efficacy, and lower score of problematic/dysfunctional coping. Primary informal carers reported a high level of HL and eHL. Carers with higher HL were more likely to report higher score of eHL.
Fauth et al. (2021) [31], USA	Aim: To evaluate a pilot Acceptance Commitment Therapy (ACT) for caregivers program, a community-based, self-guided, online adaptation of ACT. Design: Pre/post-test and four week follow-up. Population: Family caregivers. Sample size: 160 Ethics: Yes	Context: An online dementia education library. Concept: Online ACT for family caregivers on evaluating depressive symptoms, burden, stressful reactions to Behavioral and Psychological Symptoms of Dementia (BPSD), positive aspects of caregiving, and quality of life.	Online ACT showed decreased depressive symptoms, burden, and stress reactions to behavioral symptoms, and increased positive aspects of caregiving and quality of life. ACT-specific measures improved, with decreases in cognitive function and psychological inflexibility, and improvements in living according to personal values, i.e., valuing progress increased; valuing obstruction decreased.
Griffiths et al. (2018) [32], USA	Aim: Assessment of caregiver burden, caregiver competency, and frequency of Behavioral and Psychological Symptoms of Dementia (BPSD). Design: Six weeks pre/post-test. Sample population: Dementia caregivers. Sample size: 22 Ethics: Yes.	Context: Online Tele-Savvy caregiver program. Concept: Psychoeducation program on caregiver burden, caregiver competency, and frequency BPSD.	Contributed significantly to reductions in caregiver burden and depressive symptoms, a significant increase in caregiver competence, and significant reductions in the average number of BPSD and the BPSD that occur daily or more.
Kajiyama et al. (2018) [33] South America	Aim: Develop and evaluate a culturally appropriate intervention for Hispanic/Latino caregivers of persons with dementia using a structured online program Design: Six weeks pre/post-test Sample population: Dementia caregivers. Sample size: 25 Ethics: Yes	Context: Webnovela Mirela, an online Spanish Telenovela program. Concept: Webnovela Mirela, an online Spanish Telenovela psychoeducation program to teach caregivers coping with dementia caregiving.	Significant decrease in levels of stress and symptoms of depression among dementia caregivers. Technology enables support programs to reach a broader audience in a cost-effective manner.
Madden et al. (2022) [34], the UK	Aim: To increase the accessibility and efficiency of care support for families affected by dementia. Design: Participatory action research Sample population: Family caregivers Sample size: 52 Ethics: Not applicable	Context: Online video consultations Concept: Video consultation to increase access and efficiency of care support for family caregivers	Video consultations enable carers to put face to voice, focus, and connected compared to using a telephone conversation, which made them felt disconnected and unfocused. Care professionals were anxious when discussing during video consultation, due to absence of physical contact, which often served as prompts, sustained their discussions, and encouraged disclosure. Lack of facial contact led to feeling helpless and disempowerment. Inability to use therapeutic touch. Challenges to change from one task to another with video consultation. The situation was described as firefighting and confusing; frequent changes occurred regarding personal and professional info and guidelines; expressed disappointment for the initiated video consultation not being offered routinely for its innovative nature.
Poole (2020) [35], the UK	Aim: To provide accessible sources of relevant and engaging information to family carers, and to enable carers to create a space in which they could engage in peer support. Design: Cross-sectional design. Sample population: Family caregivers, care pro-fessional. Sample size: 36 Ethics: Yes.	Context: Massive Open Online Courses Survey (MOOCs). Concept: Online resource for educational and support need for family caregivers and care professionals.	The MOOC survey showed that over 90% of learners reported acquiring new skills and knowledge. Over two-thirds and three-quarters applied knowledge acquired in their daily lives and shared what they have learned with others. The peer support element enhanced learner interactions, as well as the course content, e.g., comparing experiences; offering guidance; suggesting solutions; exchanging hints and tips; providing encouragement and condolences; and empathizing with each other. The ‘safe space’ section also provided encouragement for the sharing of personal feelings regarding sensitive matters, e.g., grieving process, however, posts from participants largely influence the course content managed by the course educator to control inappropriate posts. Although developed for family caregivers, it has seen the likes of persons with dementia and care professionals advancing their knowledge in dementia care and progression.
Romero-Mas et al. (2021) [36], Spain	Aim: Describe the relation between QoL of the family caregivers of a person with Alzheimer’s disease, and their participation in a virtual community of practice (vCoP). Design: Pre/post-test quasi-experimental. Sample population: Family caregivers and health professionals Sample size: 38 with 1 drop out. Ethics: Yes.	Context: The application “Estic amb tu - I’m with you” Concept: Online social support network to evaluate QoL of family caregiver’s participation in vCoP	The application “Estic amb tu - I’m with you” showed that caregivers improve their QoL while participating in a vCoP. There was no significant difference in QoL among male and female, except for their age demographic parameter. The authors noted a negative correlation between “length of caring” and psychological and social domain of QoL. There was a difference between the relationship with the person with Alzheimer’s (spouse, offspring, and others), and overall QoL was improved through vCoP, except for the spouse parameter. Family caregivers found meaning in providing care to a loved one, feeling more useful, gaining new skills, and experience. There was no correlation between the functional deterioration of the person with Alzheimer and caregivers’ QoL. eHealth literacy impacted positively on the physical domain of the caregivers’ QoL.
Wilkerson et al. (2018) [37] USA	Aim: The study introduced Friendsourcing Web application (FPS) and examines the effects on the psychological well-being of AD caregivers. Design: Pre/post test Sample population: AD caregivers Sample size: 12 Ethics: Yes	Context: Friendsourcing Web application Concept: AD caregivers discussion on emotional and informational support via Friendsourcing Web application	A Wilcoxon signed-rank test showed that caregiving burden and perceived stress while emotional and informational support significantly reduced and increased, respectively. Subjective interviews and discussions identified: caregiving belief experience, cognitive transformation, and new behaviors learned (seeking and trying new caregiving ways/roles and applying them) from Facebook social network described as ‘’FPS’’ to reduce burden and perceived stress.

**Table 2 healthcare-11-00692-t002:** Themes and subthemes emerging after analysis.

Themes	Subthemes
Knowledge acquisition	Knowledge sharing on experience and perception, psychosocial well-being, and health literacy. Knowledge development on positive experience, psychosocial well-being, caregiver competency, strategies for self-management, and care support strategies.
Strengthening resilience capacity	Facilitators associated with strengthen resilience capacity including self-esteem; new knowledge; informal caregiver competence and females as informal caregivers

## Data Availability

Data sharing not applicable. No new data were created or analyzed in this study. Data sharing is not applicable to this article.

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
