# Peer review of "Collaborative Learning through a Virtual Community of Practice in Dementia Care Support: A Scoping Review"

_healthcare, 2023, doi:10.3390/healthcare11050692_

Round 1
Reviewer 1 Report
Thank you for this scoping review, with the aim of analysis of the vCoP, an interesting tool for dementia care support.
As suggestions for improvements to the article:
1. Introduction: well argued. They present the obligatory questions in a scoping review.
2. Materials and Methods: the search equations are in accordance with the aims of the research, but it could be clarified whether the search of reviews has been done. Please explain here why the exclusion of grey literature.
I suggest to attach the PRISMA-ScR checklist to the submission.
3. Results: in this scoping review the results have been limited, so the question of the inclusion of grey literature is once again raised.
4. Discussion: well presented.
5. Conclusions: good discussion.
Thank you very much for this article. A scoping review is usually difficult to achieve, and also in relation to the topic discussed. However, I think that the grey literature should be analysed, it would give more background to the work. I encourage the authors to consider this question as the article is of good quality and I consider it a good start to a new research topic.
Author Response
Thank you for this scoping review, with the aim of analysis of the vCoP, an interesting tool for dementia care support.
Author’s response: Thank you very much for reviewing our scoping review.
As suggestions for improvements to the article:
- Introduction: well argued. They present the obligatory questions in a scoping review. Author’s response: Thank you very much.
- Materials and Methods: the search equations are in accordance with the aims of the research, but it could be clarified whether the search of reviews has been done. Please explain here why the exclusion of grey literature.
I suggest to attach the PRISMA-ScR checklist to the submission.
Author’s response: Thank you. We did search the grey literature, added on line 116-118 but didn´t find anything. Grey literature is usually based on research, and this is a new area of research and we didn´t find any grey literature.
We have now mentioned the PRISMA-ScR in line 133-134 and added PRISMA-ScR as an appendix, Appendix A.
- Results: in this scoping review the results have been limited, so the question of the inclusion of grey literature is once again raised.
Author’s response: please, see response to Materials and Methods.
- Discussion: well presented. Author’s response: Thank you very much.
- Conclusions: good discussion. Author’s response: Thank you very much.
Thank you very much for this article. A scoping review is usually difficult to achieve, and also in relation to the topic discussed. However, I think that the grey literature should be analysed, it would give more background to the work. I encourage the authors to consider this question as the article is of good quality and I consider it a good start to a new research topic.
Author’s response: Thank you very much. Please, see response regarding grey literature in Materials and Methods.
Reviewer 2 Report
Reviewer's summary after reading the manuscript:
Research on reflective collaborative knowledge acquisition via virtual communities of practice (vCoP) was the focus of this scoping study. Research on the enabling and constraining factors of resilience capacity and knowledge acquisition through the vCoP was also identified, synthesized, and reported. PsycINFO, CINAHL, PubMed, EMBASE, Scopus, and Web of Science were searched for relevant articles. The review followed the guidelines established by Preferred Reporting Items for Systematic Reviews (PRISMA) and the meta-analyses for Scoping Reviews framework. This systematic review includes eight articles produced in English between January 2017 and February 2022. Six of them are quantitative studies, while the other two are qualitative. Quantitative descriptive summaries and qualitative theme analyses were used to synthesize the data. The first was "knowledge acquisition," while the second was "resilience building." Literature reviews offer proof that vCoP help people with dementia and their caretakers learn new things and become more resilient. As a result, it seems that vCoP may be an effective tool for enhancing dementia care. But before the notion of vCoP can be used globally, further research is required that includes less developed nations.
----------------------------------------
Dear authors, thank you for your manuscript. I enjoyed reading it. Presented are some suggestions to improve it:
(1) To improve the impact and readership of your manuscript, the authors need to clearly articulate in the Abstract and in the Introduction sections about the uniqueness or novelty of this article, and why or how it differs from other similar articles.
(2) Please substantially expand your review work, and cite more of the journal papers published, since this is a review article. There are currently only 51 citations.
(3) The references cited have adequately been formatted according to MDPI's guidelines. Just a gentle reminder: for the references, instead of formatting "by-hand", please kindly consider using the free Zotero software (https://www.zotero.org/), and select "Multidisciplinary Digital Publishing Institute" as the citation format, since there are currently 51 citations in your manuscript, and there may probably be more once you have revised the manuscript.
(4) There is a "Limitations" section (starting from line 284 until line 303) to discuss the challenges faced. That is good. However, please also consider describing how your team overcame those challenges. This would be very beneficial to the readers as they would be able to learn from your expert knowledge.
Thank you.
Author Response
Dear authors, thank you for your manuscript. I enjoyed reading it.
Author’s response: Thank you very much for reviewing our scoping review.
Presented are some suggestions to improve it:
(1) To improve the impact and readership of your manuscript, the authors need to clearly articulate in the Abstract and in the Introduction sections about the uniqueness or novelty of this article, and why or how it differs from other similar articles.
Author’s response: thank you for this comment and this is now added in the abstract, line 12-13 and in the introduction, line 75-76.
(2) Please substantially expand your review work, and cite more of the journal papers published, since this is a review article. There are currently only 51 citations.
Author’s response: thank you for this comment. Research in this area is sparse but a literature peeling added two citations to this review, see table 1, line 173-174.
(3) The references cited have adequately been formatted according to MDPI's guidelines. Just a gentle reminder: for the references, instead of formatting "by-hand", please kindly consider using the free Zotero software (https://www.zotero.org/) and select "Multidisciplinary Digital Publishing Institute" as the citation format, since there are currently 51 citations in your manuscript, and there may probably be more once you have revised the manuscript.
Author’s response: Thank you for this suggestion, we are now using EndNote.
(4) There is a "Limitations" section (starting from line 284 until line 303) to discuss the challenges faced. That is good. However, please also consider describing how your team overcame those challenges. This would be very beneficial to the readers as they would be able to learn from your expert knowledge.
Author’s response: the studies included in this scoping review were restricted to the Western world geographical aera and there is a need for studies to be conducted in other areas such as other less developed countries to open up the scope, added on line 317-320.
Thank you.
Reviewer 3 Report
The article "Collaborative knowledge acquisition through virtual Community of Practice in dementia care support: A scoping review" addresses an important issue, as dementia is a global health problem.
The article follows the rigor required for a scoping review, therefore, I emphasize that the authors correctly applied the review technique. Aspect that allowed them to answer all their questions and problems, which led to achieving the objectives of the studies.
The article is good and deserves to be published, however, I believe that the authors can improve more the discussion section of the article. I particularly recommend that authors analyze the importance of these communities of practice for the production of health knowledge and how this can be adopted by public health authorities in other countries.
Author Response
The article "Collaborative knowledge acquisition through virtual Community of Practice in dementia care support: A scoping review" addresses an important issue, as dementia is a global health problem.
The article follows the rigor required for a scoping review, therefore, I emphasize that the authors correctly applied the review technique. Aspect that allowed them to answer all their questions and problems, which led to achieving the objectives of the studies.
Author’s response: Thank you very much for reviewing our scoping review.
The article is good and deserves to be published, however, I believe that the authors can improve more the discussion section of the article. I particularly recommend that authors analyze the importance of these communities of practice for the production of health knowledge and how this can be adopted by public health authorities in other countries.
Author’s response: A section is now added to the discussion, line 299-306.
Reviewer 4 Report
Thank you for the opportunity to review this paper. This scoping review reported knowledge acquisition and strengthen resilience capacity through vCoP for persons with dementia, informal and formal caregivers. Although the work is good it has some pitfalls:
-Title. There are two themes “knowledge acquisition” and “ strengthen resilience capacity” in this review. The title only covered one of them.
-I would recommend adding search terms in the Materials and Methods section.
-figure 1. The second box under “Identification” is a little confusing. It seems like mixed the count happened before and after “Records identified with… ”. I would suggest reorganizing the second box “Unrelated records removed …”. If Identification box 1 (n=156) - Identification box 2 (n=67) = Screening box 1 (n=89), it would be easy to understand.
-table 1. I would suggest adding more columns. The aim, design, population, etc.. could be an individual column. If possible, plead to add a sample size for these studies. Table 1 only includes 6 studies instead of 8 studies which you mentioned in #161.
-table 2. I would recommend reorganizing table 2. It is not straightforward to understand which theme match which subtheme.
-The themes appeared in the aims and research questions. At the same time, the themes were from the research results. It is not logically correct. Could you please explain it?
Author Response
Thank you for the opportunity to review this paper. This scoping review reported knowledge acquisition and strengthen resilience capacity through vCoP for persons with dementia, informal and formal caregivers.
Author’s response: Thank you very much for reviewing our scoping review.
Although the work is good it has some pitfalls:
-Title. There are two themes “knowledge acquisition” and “ strengthen resilience capacity” in this review. The title only covered one of them.
Author’s response: We have now reanalyzed and changed the title.
-I would recommend adding search terms in the Materials and Methods section.
Author’s response: We tried to clarify the search terms in the manuscript, line 103-105. Adding as an appendix is not a good idea since it´s five pages. Supplementary materials can be provided upon request to the corresponding author.
-figure 1. The second box under “Identification” is a little confusing. It seems like mixed the count happened before and after “Records identified with… ”. I would suggest reorganizing the second box “Unrelated records removed …”. If Identification box 1 (n=156) - Identification box 2 (n=67) = Screening box 1 (n=89), it would be easy to understand.
Author’s response: Thank you for this comment, we have now reorganized, updated and replaced with a new figure, line 145.
-table 1. I would suggest adding more columns. The aim, design, population, etc.. could be an individual column. If possible, plead to add a sample size for these studies. Table 1 only includes 6 studies instead of 8 studies which you mentioned in #161.
Author’s response: Thank you and we are sorry for this. In the manuscript we submitted on January 13, 2023, all eight references were in Table 1 and after the editorial office made formatting changes, only six references were presented. These are now presented in the table, line 168-169. Sample size is added for each article in Table 1, line 173-174.
The table was presented in landscape orientation and is changed to portrait orientation, and we think that adding more columns would make the table harder to read.
-table 2. I would recommend reorganizing table 2. It is not straightforward to understand which theme match which subtheme
Author’s response: We have now separated the themes in separate rows to make this clearer, line 177.
-The themes appeared in the aims and research questions. At the same time, the themes were from the research results. It is not logically correct. Could you please explain it?
Author’s response: Please, look at the response to the title.
Reviewer 5 Report
The manuscript deals with the problem of how to help/form CG in the assistance of AD patients.
Methodological approache is correct, and clearly the readers could understand what kind of work was done by autors. Easily reproducible, according with authors methodology.
My only concerns deals with the "plus" that this manuscript could give.
As a minor suggestion, Authors need to clearly state (in a more easy way for readers) what is COP and vCOP.
Line 52 CoP was introduced for first time but the explanation is reported in line 57.
Author Response
The manuscript deals with the problem of how to help/form CG in the assistance of AD patients.
Methodological approache is correct, and clearly the readers could understand what kind of work was done by autors. Easily reproducible, according with authors methodology.
Author’s response: Thank you very much, and for reviewing our scoping review.
My only concerns deals with the "plus" that this manuscript could give. Author’s response: We have added a section in the discussion line 299-306 and hope to fulfill the request.
As a minor suggestion, Authors need to clearly state (in a more easy way for readers) what is COP and vCOP.
Line 52 CoP was introduced for first time but the explanation is reported in line 57.
Author’s response: Thank you for noticing this. This is now changed, and we explain CoP the first time mentioned on line 53-54 and use the abbreviation on line 59.
Round 2
Reviewer 1 Report
Thank you for the incorporation of the suggestions made. Although you have improved all those aspects mentioned, the English used in the draft needs to be improved.
Author Response
Thank you for the incorporation of the suggestions made. Although you have improved all those aspects mentioned, the English used in the draft needs to be improved.
Author’s response: Thank you very much. We have now checked the English language throughout the manuscript and made changes on lines 31-32, 40, 55, 68, 97-99, 103, 113, 158, 195-196, 200-201, 206, 210, 212, 217, 293, 306-307, 323, 326, and 334.
Reviewer 4 Report
The authors have satisfactorily addressed most of my concerns in my previous reviews. Below I have included one suggestion that could further clarify the manuscript.
-Figure 1. The relationship between “Identification” box 1 and box 2 is confusing.
It is easy to understand that “Records screened against title & abstract (n=89)” - “Records excluded (n=56)” = ”Reports sought to retrieval (n=33)”. Based on the same logic, “Identified from database (n=156)” - “Records removed before screening (n=1203) (n=67)” would be equal to “Records screened against title & abstract (n=89)”. Please search PRISMA flow chart online and modify figure 1.
Author Response
The authors have satisfactorily addressed most of my concerns in my previous reviews. Below I have included one suggestion that could further clarify the manuscript.
-Figure 1. The relationship between “Identification” box 1 and box 2 is confusing.
It is easy to understand that “Records screened against title & abstract (n=89)” - “Records excluded (n=56)” = ”Reports sought to retrieval (n=33)”. Based on the same logic, “Identified from database (n=156)” - “Records removed before screening (n=1203) (n=67)” would be equal to “Records screened against title & abstract (n=89)”. Please search PRISMA flow chart online and modify figure 1.
Author’s response: Thank for this observation. We have now modified the PRISMA flowchart to make it clearer.
Reviewer 5 Report
The quality of the paper was surely improved.
Author Response
The quality of the paper was surely improved.
Author’s response: Thank you very much for this response.